

# Pushing the limits of whole genome amplification: successful sequencing of RADseq library from a single microhymenopteran (Chalcidoidea, *Trichogramma*)

Astrid Cruaud[1], Géraldine Groussier[2], Guenaëlle Genson[1], Laure Sauné[1], Andrew Polaszek[3] and Jean-Yves Rasplus[1]

[1] CBGP, INRA, CIRAD, IRD, Montpellier SupAgro, Univ Montpellier, Montpellier, France
[2] Institut Sophia Agrobiotech, INRA, CNRS, Université Côte d'Azur, Sophia Antipolis, France
[3] Department of Life Sciences, Natural History Museum, London, United Kingdom

Corresponding author
Astrid Cruaud, astrid.cruaud@inra.fr

## ABSTRACT

A major obstacle to high-throughput genotyping of microhymenoptera is their small size. As species are difficult to discriminate, and because complexes may exist, the sequencing of a pool of specimens is hazardous. Thus, one should be able to sequence pangenomic markers (e.g., RADtags) from a single specimen. To date, whole genome amplification (WGA) prior to library construction is still a necessity as at most 10 ng of DNA can be obtained from single specimens (sometimes less). However, this amount of DNA is not compatible with manufacturer's requirements for commercial kits. Here we test the accuracy of the GenomiPhi kit V2 on *Trichogramma* wasps by comparing RAD libraries obtained from the WGA of single specimens (F0 and F1 generation, about1 ng input DNA for the WGA (0.17–2.9 ng)) and a biological amplification of genomic material (the pool of the progeny of the F1 generation). Globally, we found that 99% of the examined loci (up to 48,189 for one of the crosses, 109 bp each) were compatible with the mode of reproduction of the studied model (haplodiploidy) and Mendelian inheritance of alleles. The remaining 1% (0.01% of the analysed nucleotides) could represent WGA bias or other experimental/analytical bias. This study shows that the multiple displacement amplification method on which the GenomiPhi kit relies, could also be of great help for the high-throughput genotyping of microhymenoptera used for biological control, or other organisms from which only a very small amount of DNA can be extracted, such as human disease vectors (e.g., sandflies, fleas, ticks etc.).

# INTRODUCTION

Parasitoid wasps (especially Chalcidoidea; *Heraty et al., 2013*) are increasingly used as biocontrol agents of many crop pests to reduce pesticide use (*Austin & Dowton, 2000*). Among them, minute wasps of the genus *Trichogramma* (210 species worldwide, 40 in

Europe), which develop within the eggs of 200 species of moths damaging crops (e.g., apple, banana, grape, maize, pine, tomato; *Consoli, Parra & Zucchi, 2010*; *Polaszek et al., 2012*) are the most commercialized worldwide.

It is acknowledged that successful and safe biological control depends on accurate genetic and phenotypic characterization of the strains released. Furthermore, host preferences and the potential of strains to hybridize with each other, or with native species, should be carefully studied. This is critical to avoid non-target effects such as gene introgression with indigenous species (*Van Driesche & Hoddle, 2016*). However, probably because most species of chalcids are minute wasps (less than a few millimetres long, and as little as 0.16 mm) and are difficult to identify to species by non-specialists, strains are often released without in-depth characterization.

RADseq, the sequencing of hundreds of thousands of DNA fragments flanking restriction sites (*Miller et al., 2007*) has been successfully used for population genetics or phylogeography (*Emerson et al., 2010*) to infer relationships between closely (*Jones et al., 2013*; *Nadeau et al., 2013*; *Wagner et al., 2013*) or more distantly (*Cruaud et al., 2014*; *Hipp et al., 2014*) related species, to detect hybridization processes (*Eaton & Ree, 2013*; *Hohenlohe et al., 2011*), to identify markers under selection and detect genes that are candidates for phenotype evolution (*Hohenlohe et al., 2010*), or to better understand the genomic architecture of reproductive isolation (*Gagnaire et al., 2013*). Thus, sequencing RAD markers appears relevant for in-depth characterisation of *Trichogramma* species and strains used in biocontrol.

A major obstacle to RAD sequencing of oophagous parasitoids is their small size. Ideally, one should be able to sequence RAD markers from a single specimen. Indeed, species complexes exist that are difficult to identify based on morphology only (*Al Khatib et al., 2014*; *Kenyon et al., 2015*; *Mottern & Heraty, 2014*), which makes sequencing of a pool of specimens risky. However, to date, the DNA amount obtained from single specimens is not sufficient enough to build a RADseq library. Usually, for minute specimens, at most 10 ng of DNA is obtained (often less), whereas about 150 ng of DNA is required to build a RADseq library from our experience, and much more can be required by private companies. Performing whole genome amplification (WGA) prior to library construction is thus a necessity. So far a few studies have formally examined the accuracy of WGA methods, mostly on human DNA and either a few loci (*Hosono et al., 2003*; *Lovmar et al., 2003*; *Sun et al., 2005*) or a higher number of SNPs and loci but always with 10 ng or more input DNA (*Abulencia et al., 2006*; *Barker et al., 2004*; *Blair, Campbell & Yoder, 2015*; *El Sharawy et al., 2012*; *Paez et al., 2004*; *Pinard et al., 2006*). All studies have concluded that the multiple displacement amplification method, MDA, (*Dean et al., 2002*; *Lasken, 2009*), which relies on isothermal DNA amplification using a high-fidelity polymerase bacteriophage phi29; (*Paez et al., 2004*) and random hexamer primers to decrease amplification bias and increase product size, is among the most accurate.

So far, only one study has quantified sequence bias that might result from WGA prior to double-digest RAD sequencing, ddRADseq; (*Peterson et al., 2012*); a variant of RADseq that uses two restriction enzymes to cut DNA instead of one enzyme and a DNA shearing system. In their study, *Blair, Campbell & Yoder (2015)* use the Qiagen REPLI-g Mini Kit and

100 ng of input DNA (as requested by the kit) extracted from liver samples of specimens of the grey mouse lemur (*Microcebus murinus*). They conclude that the kit does not introduce bias for (i) SNP calling as compared to what is obtained from native DNA of the same samples or (ii) genome coverage as compared to the published genome of *M. murinus*. Here we test the accuracy of the GE Healthcare Life Sciences[TM] illustra[TM] GenomiPhi V2 for the WGA of single *Trichogramma* wasps prior to RADseq library construction. As for the REPLI-g Mini Kit, WGA is performed using the MDA. However, the GenomiPhi kit requires 10 times less DNA (1 µl of input DNA at 10 ng/ µl) but still more than what can be extracted from single *Trichogramma* wasps. As a consequence, we had to push the limits of the kit, increasing the risk of inconsistent or unrepresentative amplification of the genome. To test the accuracy of the GenomiPhi kit in these challenging conditions we took advantage of the mode of reproduction of *Trichogramma* wasps (arrhenotokous parthenogenesis; i.e., females develop from fertilized eggs and are diploid, while males develop from unfertilized eggs and are haploid). We compared the number and sequences of RAD tags obtained from the WGA of single individuals from the F0 and F1 generation with the number and sequences of RAD tags obtained from the pool of their progeny (F2 generation) (Fig. 1). Thus, we compared RAD libraries obtained from a technical / artificial amplification (WGA) and a biological/natural amplification (pool of specimens).

## MATERIALS AND METHODS

### Sampling and experimental design

The species *Trichogramma brassicae* Bezdenko, 1968 was used as model system. Specimens were taken from the strain collection hosted by the Biological Resource Centre ''Egg Parasitoid Collection'' (EP-Coll, Sophia-Antipolis, France) (*Marchand et al., 2017*) and confirmed as *T. brassicae* by AP and JYR using morphological characters detailed by *Pino et al. (2013)*, especially the ratio between the length of the longest antennal seta and maximum antennal width. Voucher specimens from this study are deposited permanently at the Natural History Museum, London. Male (haploid)/female (diploid) pairs were placed in glass tubes and left free to mate (1 pair per tube, Fig. 1). Droplets of honey were provided as food, and eggs of *Ephestia kuehniella* (Pyralidae) were used as hosts. F0 females and males were killed in 70% ethanol before emergence of the F1 generation. Emerging females of the F1 generation were kept separated from males (no mating) and reared in new glass tubes (1 virgin female per tube). Again, droplets of honey were provided as food and eggs of *E. kuehniella* were used as hosts. F1 females were killed before the emergence of the F2 generation, which was composed only of males, as the reproductive strategy of *T. brassicae* is arrhenotokous parthenogenesis. Ten parental crosses (F0 male × female) were attempted. For each cross, all F2 males were pooled prior to DNA extraction. For each cross, a WGA was performed prior to RAD library construction on the F0 female, the F0 male and one F1 female, while all F2 males were pooled prior to DNA extraction and the resulting DNA was used directly as input for RAD library construction without WGA. Thus, the F2 generation was used as a negative control. In *T. brassicae*, females develop from fertilized eggs and are diploid, while males develop from unfertilized eggs

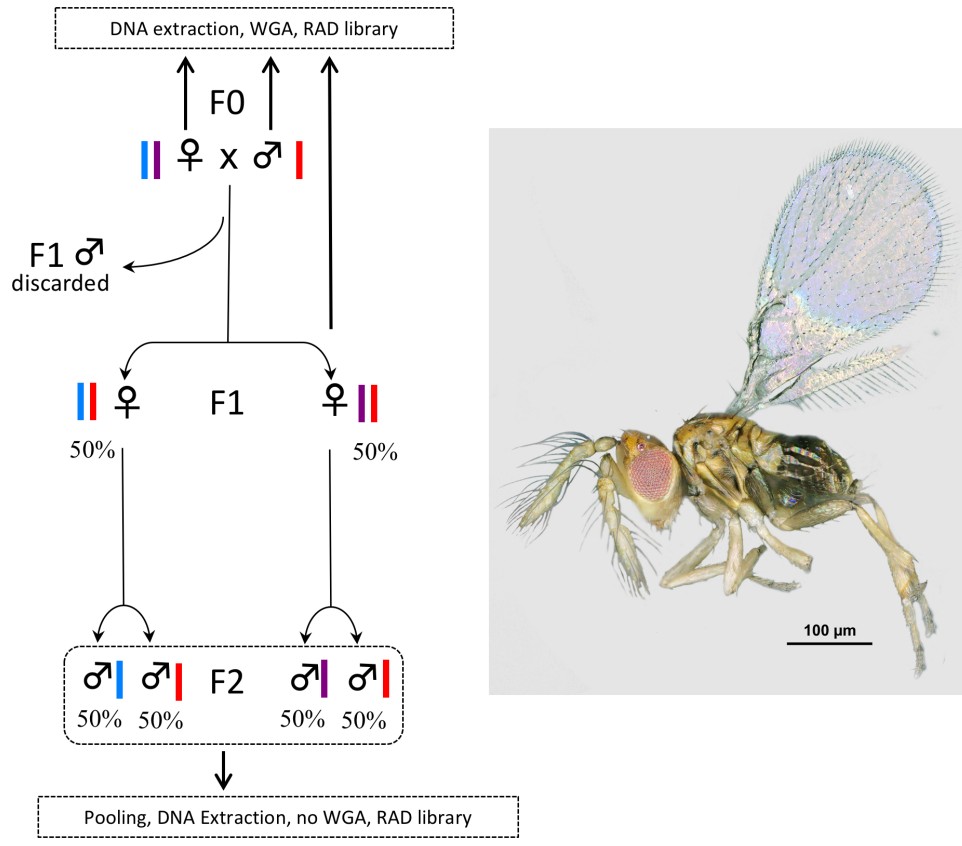

**Figure 1 Experimental setup.** In *Trichogramma*, females develop from fertilized eggs and are diploid, while males develop from unfertilized eggs and are haploid. In this figure, the coloured bars close to the female and male symbols represent alleles. The breeding experiment was as follows: (1) A female/male pair (F0 generation) was left free to mate in a glass tube (one pair per tube). Sterilized eggs of *E. kuehniella* (Pyralidae) were provided for oviposition. (2) Only females of the F1 generation were kept, males were discarded. Virgin F1 females were isolated into glass tubes (one female per tube). Again, sterilized eggs of *E. kuehniella* (Pyralidae) were provided for oviposition. (3) All F2 males were kept. This breeding experiment was replicated ten times. For clarity and to provide allele frequency predicted by Mendel's laws of inheritance, only two F1 females are represented on the figure. Similarly, only two F2 males are represented for each F1 female. For each replicate, a WGA was performed prior to RAD library construction on the F0 female, the F0 male and one F1 female, while all F2 males were pooled prior to DNA extraction and the resulting DNA was directly used as input for RAD library construction without WGA. Thus, the F2 generation was used as a negative control. Photo *T. brassicae* male ©J-Y Rasplus.

and are haploid. We took advantage of this reproductive strategy to test for potential bias introduced by WGA. Indeed, with such a reproductive strategy, expected results are as follows: for all RAD tags (i) F0 males should be haploid, (ii) F0/F1 females should be diploid and either homozygous or heterozygous. We also expect that most RAD tags will follow *Mendelian* laws of inheritance (Fig. 1). We thus expect compatibility of parental and offspring genotypes from the F0 to the F2 generation. Hence, for example, (i) all SNPs found in the pool of F2 males should be present in the F0 generation; (ii) F1 females should be heterozygous when their parents possess different alleles.

## DNA extraction and whole genome amplification

DNA extraction was performed with the Qiagen DNeasy 96 Blood & Tissue Kit, following manufacturer protocol with the following modifications to increase DNA yield: two successive elutions (50 μL each) were performed with heated buffer AE (55 °C) and an incubation step of 15 min followed by plate centrifugation (6,000 rpm for 2 min).

DNA was quantified with a Qubit® 2.0 Fluorometer (Invitrogen, Carlsbad, CA, USA). To stay as close as possible to the recommended amount/volume of DNA listed in the GenomiPhi protocol (1 μl DNA input at 10 ng/μl), ethanol precipitation of DNA was performed prior to WGA. 1/10 volume of sodium acetate 3 M pH 5.2 was added to the extract. Two volumes of cooled absolute ethanol were then added to the mix which was then incubated at −20 °C overnight. The mix was then centrifuged (30 min, 13,000 rpm, 4 °C) and the pellet was washed with 500 μl of cooled 70% ethanol. After another centrifugation (15 min, 13,000 rpm, 4 °C), the pellet was dried at room temperature and resuspended in 4 μl of sterile molecular biology ultrapure water, as a total resuspension of the pellet would not have been obtained in a smaller volume. Concentrated DNA was quantified with a Qubit® 2.0 Fluorometer (Invitrogen, Carlsbad, CA, USA). DNA extracts were then subjected to WGA using the GenomiPhi™ V2 DNA Amplification kit (GE Healthcare, Chicago, IL, USA) with 1ul of concentrated DNA used as input. The resulting DNA was quantified with a Qubit® 2.0 Fluorometer (Invitrogen, Carlsbad, CA, USA).

## RADseq library construction

Library construction followed *Baird et al. (2008)* and *Etter et al. (2011)* with modifications detailed in *Cruaud et al. (2014)*. The *PstI* enzyme was used as the cutter. The number of expected cut sites was estimated with an *in silico* digestion of the genome of *T. pretiosum* (assembly Tpre_1.0, 196 Mb) using a custom script. The quantity of P1 adapters (100 nM) to be added to saturate restriction sites (result = 3 uL) as well as the optimal time for DNA sonication on a Covaris S220 ultrasonicator to obtain fragments of 300–600 bp (results = duty cycle 10%, intensity 5, cycles/burst 200, duration 70 s) that are both specific to the studied group were evaluated in a preliminary experiment. Briefly, RADseq libraries were built as follows: The experiment to test the accuracy of WGA for RADseq of microhymenoptera is part of a larger project that aims to resolve the phylogenetic relationships of European *Trichogramma* wasps. Thus, more samples ($N = 40$) than those used to answer our technical question were included in the library. About 250 ng of input DNA was used for each sample. After digestion with *PstI*, samples were individually tagged with barcoded P1 adapters. Samples were then pooled eight by eight. Five pools of eight samples were obtained. The DNA of each pool was sheared using a Covaris ultrasonicator. After size selection on gel (300–600 bp), end repair and 3′-end adenylation, each pool was tagged with a different barcoded P2 adapter. A PCR enrichment step was then performed. For each pool, five independent PCR reactions were used to increase fragment diversity (10 ng input DNA, 13 cycles, NEB Phusion High-Fidelity PCR Master Mix). The five PCR products were pooled and purified with Ampure beads (Beckman Coulter, Brea, CA, USA). The resulting enriched libraries ($N = 8$) were quantified with Qubit, an Agilent Bioanalizer and qPCR with the ≪ Library Quantification Kit - Illumina/Universal ≫ from

KAPA (KK4824) and pooled at equimolar ratio prior to sequencing. 2*125 nt paired-end sequencing was performed at MGX-Montpellier GenomiX on one lane of an Illumina HiSeq 2500 flow cell.

### Data analysis

Cleaning of raw data was performed with the wrapper RADIS (*Cruaud et al., 2016*) that relies on Stacks (*Catchen et al., 2013*; *Catchen et al., 2011*) for demultiplexing of data and removing PCR duplicates. Data analysis was performed with Stacks v1.46. Individual loci were built using *ustacks* ($m = 3$; $M = 1$; $N = 2$; with the removal (r) and deleveraging (d) algorithms enabled). Each cross was analysed separately. Catalogs of loci were built with *cstacks* ($n = 2$). First, a catalogue grouping F1 females and their progeny was built. Then a catalogue grouping samples from the F0, F1 and F2 generation was built. *sstacks* was used to map individual loci to the catalogue. *rxstacks* was then used to correct genotype and haplotype calls: (i) Loci for which at least 50% of the samples (when a pair composed of one F1 female and a pool of F2 males was analysed) or 25% of the samples (when F0, F1 and F2 were analysed together) had a confounded match to the catalogue were removed; (ii) excess haplotypes were pruned; (iii) SNPs were recalled after removal of possible sequencing errors using the bounded SNP model (–bound_high 0.1), and (iv) loci with an average log likelihood less than $-10.0$ were discarded. After this filtering step, *cstacks* and *sstacks* were rerun. The program *populations* was then used to compare the RAD tags obtained with or without WGA (parsing of the *haplotypes.tsv* and *populations.log* files). Loci were kept only if (i) they had a minimum stack depth of 10 and (ii) all samples had a sequence. Analyses were performed on the Genotoul Cluster (INRA, Toulouse, France).

## RESULTS

On the ten attempted crosses, the number of F1 females varied from 0 to 38. Only three parental crosses produced enough F2 males ($N > 100$) to get a sufficient amount of DNA for RADseq library construction without WGA. Consequently, RADseq libraries were constructed only on these crosses. DNA extraction of one-third of the tested specimens provided an amount of DNA that was below the detection limit of the Qubit (Table 1). WGA was not attempted on these specimens. For other specimens, the average amount of DNA obtained with the Qiagen kit was 10.4 ng (min = 6.2–max = 13.9) (Table 1). After DNA re-concentration, the average DNA quantity used as input for the WGA was about 1.0 ng (0.17–2.9 ng). On average, 947.5 ng of DNA was obtained with the WGA (226–2,393 ng).

*In silico* digestion of the genome of *T. pretiosum* revealed 59,433 *PstI* cut sites (i.e., 118,866 tags). An average of 2*3,757,867 reads (109 bp) was obtained for the different samples after quality filtering, demultiplexing and removal of PCR clones (Table 1). Two F1 females (TRIC00027_1103 and TRIC00027_3103) were represented by much less reads than other samples (595,204 and 1,991,305 respectively). The number of tags recovered by *ustacks* and *cstacks* varied, but was comparable among the samples and in line with the predictions made on the genome of *T. pretiosum* when these two females were excluded from calculation (average number of ustacks tags = 132,787; average number of cstacks tags = 128,293, Table 1).

Cruaud et al. (2018), *PeerJ*, DOI 10.7717/peerj.5640

**Table 1  Extraction, whole genome amplification and sequencing results.**

| Cross # | Sample code | Description | qDNA obtained after extraction (ng) | qDNA used for WGA (ng) | qDNA obtained after WGA (ng) | Input DNA used for RAD library (ng) | Nb of demultiplexed reads[a] (forward only) | Nb of cleaned reads[b] (forward only) | *ustacks*: Nb of loci | *ustacks*: Nb of loci[c] |
|---|---|---|---|---|---|---|---|---|---|---|
| 1 | TRIC00027_2101 | Male (F0), haploid, WGA | 11.5 | 1.5 | 500 | 169.0 | 6,774,680 | 5,173,711 | 136,623 | 130,060 |
| 1 | TRIC00027_2102 | Female (F0), diploid, WGA | 10.6 | 0.39 | 1,048 | 203.3 | 4,073,370 | 3,179,891 | 128,212 | 122,860 |
| 1 | TRIC00027_2103 | Female (F1), diploid, WGA | 6.20 | 0.35 | 2,393 | 281.2 | 4,597,505 | 3,566,986 | 130,565 | 124,845 |
| 1 | TRIC00027_2199 | Pool of haploid males (F2) ($n = 933$), no WGA | 735.4 | N.A. | N.A. | 269.0 | 4,818,385 | 3,745,752 | 127,709 | 125,047 |
| 2 | TRIC00027_1101 | Male (F0), haploid, WGA | Too low | N.A. | N.A. | N.A. | N.A. | N.A. | N.A. | N.A. |
| 2 | TRIC00027_1102 | Female (F0), diploid, WGA | Too low | N.A. | N.A. | N.A. | N.A. | N.A. | N.A. | N.A. |
| 2 | TRIC00027_1103 | Female (F1), diploid, WGA | 9.4 | 0.17 | 1128 | 164.6 | 774,450 | 595,204 | 43,763 | 42,062 |
| 2 | TRIC00027_1199 | Pool of haploid males (F2) ($n = 229$), no WGA | 359.6 | N.A. | N.A. | 270.6 | 5,878,301 | 4,437,984 | 127,380 | 125,302 |
| 3 | TRIC00027_3101 | Male (F0), haploid, WGA | Too low | N.A. | N.A. | N.A. | N.A. | N.A. | N.A. | N.A. |
| 3 | TRIC00027_3102 | Female (F0), diploid, WGA | 13.9 | 2.9 | 390 | 247.7 | 7,006,980 | 5,365,499 | 147,718 | 140,690 |
| 3 | TRIC00027_3103 | Female (F1), diploid, WGA | 10.7 | 0.43 | 226 | 137.3 | 2,616,330 | 1,991,305 | 93,606 | 89,543 |
| 3 | TRIC00027_3199 | Pool of haploid males (F2) ($n = 1,415$), no WGA | 670.7 | N.A. | N.A. | 228.4 | 7,510,904 | 5,764,475 | 131,267 | 129,249 |

**Notes.**

[a] Reads obtained after demultiplexing and quality filtering with *process_radtags*.

[b] Reads obtained after removal of PCR clones (input reads for the *ustacks* step).

[c] One catalog was built for each cross.

The comparison of the loci obtained after filtering steps with *rxstacks* and *populations* revealed that, on average, 97.6% of the loci were homozygous and identical in F1 females and the pool of F2 males (min = 96.8% - max = 98.2%, Table 2). On average, 0.7% (0.3%–1.3%) of the loci were heterozygous and identical in F1 females and the pool of F2 males. Thus, there was a 98.3% (97.4%–99.0%) exact match between the loci of the F1 females (whose DNA was amplified with WGA) and the whole progeny of the F1 generation (pool of males whose DNA was not amplified with WGA). Between 1.0 and 2.6% of the loci were not identical between F1 females and the pool of F2 males (Table 2). A careful inspection of the haplotypes revealed that about 60% of these differences could be explained by the experimental setup (Table 3). Indeed, RAD tags of a single female of the F1 generation are compared with RAD tags of the whole progeny of the F1 generation (Fig. 1). Globally, 99.3% (98.9–99.6%) of the shared loci were either identical or displayed differences that could be explained by the experimental setup. The first cross was used to check in detail the overall compatibility of the genotypes from the parental generation (F0) with the whole progeny of the F1 generation (Table 4). A total of 98.8% of the 32,913 loci shared by the four samples displayed haplotypes consistent with experimental setup and *Mendelian* laws of inheritance (97.3% of the loci were homozygous and identical between samples). SNPs observed in 385 loci (which represent 1.2% of the loci and 0.01% of the analysed nucleotides) were neither compatible with the mode of reproduction of the studied model (haplodiploidy) nor Mendel's laws. Considering the haplotype observed in the pool of F2 males as a reference, questionable SNPs could be categorized into five categories as listed in Table 5. In 90% of the situations, SNPs found either in the F0 male (21%), the F0 female (32.5%) or the F1 female (36.6%) were incompatible with haplodiploidy or with Mendelian inheritance of alleles. 96 cases (about 25.0%) represented situations where one allele was missing for the F0 female or the F1 female to fit with Mendelian inheritance of alleles (possible cases of allele drop-out).

## DISCUSSION

Here we compare RAD libraries obtained from a technical/artificial amplification of DNA (WGA of single specimen of micro-hymenoptera, F0 and F1 generations) and a biological/natural amplification (pool of the progeny of the F1 generation). We push the limits of the kit used for the WGA (GenomiPhi) by using 90% less DNA (about 1.0 ng) than the required amount specified on the manufacturer's protocol (10 ng). Globally, we show that 99% of the examined loci (up to 48,189; 109 bp each) were compatible with haplodiploidy and either identical among specimens or compatible with Mendelian inheritance of alleles. These results are consistent with observations by *Blair, Campbell & Yoder (2015)* who used the Qiagen REPLI-g Mini Kit and 100 ng of input DNA and showed that SNP calling between ddRAD libraries from native and amplified DNA presented a >98% match (up to 11,309 loci examined). They are also in agreement with older studies that attempted to quantify bias induced by multiple displacement amplification method (MDA) on which the GenomiPhi kit relies (>99% match; *Barker et al., 2004*; *El Sharawy et al., 2012*; *Paez et al., 2004*), though with more input DNA (10 ng).

Cruaud et al. (2018), *PeerJ*, DOI 10.7717/peerj.5640

**Table 2 Pairwise comparison of loci obtained for females of the F1 generation and pools of males of the F2 generation.** Analysed loci have been first corrected by *rxstacks* for genotype and haplotype calls and filtered with *populations*. Only loci that were present in the two samples with a stack depth of 10 were kept.

| Pair of compared samples | Nb of shared loci | Percentage of identical loci (homozygous) | Percentage of identical loci (heterozygous) | Percentage of loci with differences possibly explained by the experimental setup | Percentage of loci with differences not explained by the experimental setup |
|---|---|---|---|---|---|
| In cross #1 | 48,189 | 97.7 | 1.3 | 0.6 | 0.4 |
| F1 Female *versus* pool of F2 males | | Total percentage of identical loci 99.0 | | Total percentage of loci with differences 1.0 | |
| In cross #2 | 5,184 | 96.8 | 0.6 | 1.5 | 1.1 |
| F1 Female *versus* pool of F2 males | | Total percentage of identical loci 97.4 | | Total percentage of loci with differences 2.6 | |
| In cross #3 | 20,095 | 98.2 | 0.3 | 0.9 | 0.6 |
| F1 Female F1 *versus* pool of F2 males | | Total percentage of identical loci 98.5 | | Total percentage of loci with differences 1.5 | |

**Table 3  Summary of differences observed between F1 females and the whole progeny of the F1 generation (pool of F2 males).** For clarity, alleles are represented with capital letters A, B and C.

| | F1 female | Pool of F2 males |
|---|---|---|
| Differences explained by experimental setup (60%) | 1 allele (A) | 2 alleles (A, B) |
| | 2 alleles (A, B) | 3 alleles (A, B, C) |
| Differences not explained by experimental setup (40%) | 1 allele (A) | 1 allele (B) |
| | 2 alleles (A, B) | 1 allele (A) |
| | 2 alleles (A, B) | 1 allele (C) |
| | 1 allele (C) | 2 alleles (A, B) |
| | 2 alleles (A, C) | 2 alleles (A, B) |

**Table 4  Comparison of loci obtained for the first crossing experiment.** Analysed loci have been first corrected by *rxstacks* for genotype and haplotype calls and filtered with *populations*. Only loci that were present in the four samples with a stack depth of 10 were kept.

| Compared samples | Nb of shared loci | Percentage of identical loci (homozygous) | Percentage of loci consistent with the experimental setup and Mendelian inheritance of alleles | Percentage of loci not consistent with the experimental setup and Mendelian inheritance of alleles |
|---|---|---|---|---|
| In Cross #1<br>- F0 female<br>- F0 male<br>- one F1 female<br>- pool of F2 males (i.e., progeny of the F1 generation) | 32,913 | 97.3 | 98.8 | 1.2 |

**Table 5  Categories of SNPs not compatible with the mode of reproduction of the studied model (haplodiploidy) or Mendel's laws of inheritance and number of occurrences of each case.** The different situations are illustrated by examples taken from the analysis of the 385 questionable SNPs.

| Description | F0 Male | F0 Female | F1 Female | Pool of F2 males | Occurrences |
|---|---|---|---|---|---|
| F0 Male incompatible | **A/G** | G | G | G | 81 (21.04% of the problematic SNPs; 0.002% of the analysed nt) |
| | **TG** | GA | GA | GA | |
| F0 Female incompatible | C | C/**T** | C | C | 125 (32.47% of the problematic SNPs; 0.003% of the analysed nt) |
| | A | **A** | A/G | A/G | |
| | G | **A** | G | G | |
| F1 Female incompatible | C | C | **A**/C | C | 141 (36.62% of the problematic SNPs; 0.004% of the analysed nt) |
| | T | C/T | **C** | C/T | |
| | GG | AG/GG | GG/**GT** | AG/GG | |
| | T | A | **A** | A/T | |
| Pool of F2 males incompatible | T | T | T | **C**/T | 29 (7.53%; of the problematic SNPs; 0.0008% of the analysed nt) |
| | A | A/G | A/G | A | |
| | AA | AA | AA | **CC** | |
| | C | T | C/T | **C** | |
| Combination of the different situations | **C/G** | C/G | C/G | **C** | 9 (2.34%; of the problematic SNPs; 0.0002% of the analysed nt) |

With the exception of two samples, for which the construction of the library seems to have failed (far fewer reads were obtained), comparable numbers of tags were obtained. This indicates that the coverage of the genome is the same regardless whether native or amplified DNA is used as suggested by previous studies on the potential bias induced by MDA (*Abulencia et al., 2006*; *Blair, Campbell & Yoder, 2015*; *Paez et al., 2004*). Studies have suggested that WGA may induce allele dropout especially when the starting amount of DNA is low (<1 ng) (*Handyside et al., 2004*; *Lovmar et al., 2003*; *Lovmar & Syvänen, 2006*; *Sun et al., 2005*). *El Sharawy et al. (2012)* and *Blair, Campbell & Yoder (2015)* concluded that MDA had no significant effect on levels of homozygosity. Here about 1% of the loci retained by our analytical pipeline (i.e., about 0.01% of the examined nucleotides) presented problematic SNPs that were neither compatible with the biology of *Trichogramma* wasps nor Mendelian inheritance of alleles. 0.3% of the SNPs were possible cases of allele drop-out (one allele was missing for the F0 female or the F1 female to fit with Mendel's laws). A larger sampling would be required to examine these few problematic SNPs in more detail. Here, a correlation may exist between the number of problematic SNPs and the quantity of input DNA used for the WGA (less bias in haploid F0 male (1.5 ng; 81 problematic SNPs; 0.002% of the examined nucleotides) as compared to F0 female (0.39 ng; 125; 0.003%) and F1 female (0.35 ng; 141; 0.004%)) but no definite conclusion can be drawn. It is noteworthy that if these problematic SNPs can indeed result from bias caused by WGA, other explanations are possible (competition between fragments for ligation of P1 adapters, mutation during enrichment PCR, sequencing error). Indeed, although they are less frequent, bias are also observed in the pool of F2 males (29 problematic SNPs; 0.0002% of the examined nucleotides). Finally, some improvements could be made to our protocol. Extraction failed for a third of our specimens (especially single males that are much smaller than females). Here we used the Qiagen kit 96-well-plate format in order to facilitate the processing of many specimens simultaneously. However, especially for valuable specimens, DNA yield could be increased with the spin-column format, as higher centrifuge speed could be used. Furthermore, for projects that aim to target a high number of specimens, re-concentration on SPRI beads may be used instead of using ethanol precipitation of DNA. Indeed, such methods are compatible with robotic sample preparation. However, while DNA yield could be better, working with very low amounts of buffer to resuspend DNA could be troublesome.

## CONCLUSION

In this study, we pushed the limits of the GenomiPhi kit V2 and successfully built RADseq libraries from single micro-wasps (*T. brassicae*). Globally, we found that about 99% of the examined loci (up to 48,189; 109 bp each) were compatible with the mode of reproduction of the studied model (haplodiploidy) and/or Mendelian inheritance of alleles. The remaining 1% (about 0.01% of the analysed nucleotides) could represent WGA bias or other experimental/analytical bias. It is noteworthy that the GenomiPhi kit V2 (and the new GenomiPhi kit V3) are affordable and easy to use by most laboratories, which is an

important point to consider given the increasing demand for the genomic characterisation of parasitoids used in biocontrol programs or other disease-transmitting micro-arthropods (e.g., sand flies, fleas, ticks etc.).

## ACKNOWLEDGEMENTS

We are grateful to Montpellier GenomiX (Montpellier, France) for sequencing of the library and to the Genotoul bioinformatics platform Toulouse Midi-Pyrenees for providing computing resources. We thank two anonymous referees for their careful reading and valuable suggestions to improve our manuscript.

### Funding

This work was funded by the ANR project TriPTIC (ANR-14-CE18-0002). There was no additional external funding received for this study. The funders had no role in study design, data collection and analysis, decision to publish, or preparation of the manuscript.

### Grant Disclosures

The following grant information was disclosed by the authors:
ANR project TriPTIC: ANR-14-CE18-0002.

### Competing Interests

The authors declare there are no competing interests.

### Author Contributions

- Astrid Cruaud conceived and designed the experiments, analyzed the data, prepared figures and/or tables, authored or reviewed drafts of the paper, approved the final draft.
- Géraldine Groussier performed the experiments, contributed reagents/materials/analysis tools, approved the final draft.
- Guenaëlle Genson performed the experiments, approved the final draft.
- Laure Sauné performed the experiments, approved the final draft.
- Andrew Polaszek authored or reviewed drafts of the paper, approved the final draft.
- Jean-Yves Rasplus conceived and designed the experiments, analyzed the data, prepared figures and/or tables, authored or reviewed drafts of the paper, approved the final draft.

### Data Availability

Cleaned reads are available as a NCBI Sequence Read Archive (SRP136713).

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
