# Peer review of "Pushing the limits of whole genome amplification: successful sequencing of RADseq library from a single microhymenopteran (Chalcidoidea, Trichogramma)"

_PeerJ, doi:10.7717/peerj.5640_

## Round 0.1 · original submission · Major Revisions

I now have responses from all 3 referees on your submission, and they agree that your manuscript may become acceptable for publication pending a major revision to clarify the experimental methods and justify the sample size for the study. It is simple enough to perform a power analysis and it seems that it would be valuable to the referees who questioned whether an N of 3 is sufficient. The referee who is most familiar with the technical aspects of this study was unable to follow your experimental design and is therefore unconvinced by your study. This is the most important statement to me in regards to your revision - if expert referees are unable to evaluate your study because of lack of detail or clarity of the experimental design, then the study is not acceptable for publication. I intend to return your manuscript to this most critical referee upon revision based on the feedback you have received in these reviews.

Reviewer 1 ·

Basic reporting

This study uses the GenomiPhi kit (V2) to test for potential biases introduced through whole genome amplification (WGA) followed by RADseq. The authors were specifically interested in testing the potential utility for minute wasps that generally yield extracts with DNA quantities too low for standard RADseq library preparation. These wasps are often used as biological control agents, so proper genetic characterization is necessary. A relatively complex crossing experiment is performed in which WGA is performed on both F0 and F1 flies. RADtags are then compared to those produced from pooling F2 males and generating standard RAD libraries without WGA. The authors analyze their data using the standard Stacks pipeline and find no substantial bias in WGA samples. These conclusions are based off of the mode of reproduction of their study species and Mendelian assumptions. These results confirm previous studies and suggest that WGA through MDA may be suitable for next-generation sequencing of specimens with low DNA quantity.

Although I found the study interesting, especially as it is the first to examine the potential utility of WGA + RADseq with very low DNA quantity, I am unable to recommend the manuscript for publication in its present form. One major concern is with some of the writing, which was difficult to follow. For example, I would recommend using the term ‘70% ethanol’ versus ‘ethanol 70%.’ Along similar lines, use ‘F1 females’ instead of ‘females F1.’ Lines ~190-200 are quite difficult to follow. If not done yet, I would recommend editing by a native English speaker to improve clarity. If the authors can address the experimental design issues and quality of the writing I am sure this will make a valuable contribution to the literature.

Experimental design

My primary concern lies with the experimental design, which I found to be quite confusing throughout the paper. As someone who does not work on parasitoid wasps, I would have liked to see a bit more discussion about breeding systems in the species and how this mode of reproduction lends itself to being tested using the techniques the authors use. Figure 1 essentially has no figure legend, thus it is a bit difficult to understand how the breeding experiment was performed. What do the colored bars represent? What does the side arrow branching off of the F0 cross represent? If a reader cannot fully comprehend the experimental setup, all results and potential implications are meaningless. My recommendation is to modify Fig. 1, include an informative legend, and elaborate on the experimental setup in the Methods

Validity of the findings

The authors conclude that WGA through the GenomiPhi V2 kit combined with RADseq may be useful to study genomic patterns and processes in specimens with DNA quantities too low for standard RADseq protocols. These are potentially interesting findings as no study to date has tested for potential bias in WGA and RAD sequencing when starting from very low DNA quantities. Unfortunately, I remain a bit skeptical about the findings for two primary reasons: (1) The methods/experimental design is a bit difficult to follow. In the revision I would highly recommend modifying Figure 1 and including an informative legend. I would also elaborate on the study system in the Methods and how it can be used to test for potential biases in WGA. (2) It appears that the majority of the conclusions are based on low sample sizes. If I understand correctly, only three pools of F2 males were used for the comparisons. Can the authors justify this sample size?

Additional comments

Line 63: There should be a citation here.

Line 87: Make it clear early on what you are comparing. Are you looking at both coverage of RAD loci and individual SNPs?

Line 97: If identification to species is so difficult, how was this done in the study?

Line 139: Five pools consisting of what? How many/which samples per pool? Were RADtags amplified through PCR? I assume so, but this isn’t stated. Again, I feel that a lot of important information was omitted.

Line 164: Only three crosses had enough F2 males for pooled library preparation? So essentially you have a sample size of three? Do you think this is high enough to make a recommendation/conclusion?

Line 204 and throughout: What exactly is ‘a Mendelian inheritance?’

Line 248: So you did perform PCR?

Line 250: This is not a proper sentence.

Reviewer 2 ·

Basic reporting

I recommend publication once a more coherent draft is submitted. There are multiple instances in the article that would benefit from a thorough review. For instance

Overall Comments/examples of awkwardly worded sentences/ and statements (not complete, just representative):
Please re-think use of “ca.” and possibly remove/reword. Very Distracting
Second Paragraph (lines 41-56) very important but awkwardly worded
Line 101-105 confusing sequence explain from F0 to F2
Line 122 Consistency Whole Genome Amplification/WGA
Line 164: leaded?
Line 167: that stand?
Line 171: In average?
Line 171: (226 - 2393)? Is this ng?
Line 182: In average?

Tables:
The tables should be self-explanatory… please expand

Experimental design

Well developed assessment that should be published once it is made more coherent. The experimental design and technical aspects of the work are quite in depth and interesting, but the writing distracts from this.

Validity of the findings

The assessment is novel and the analysis robust. The Conclusions drawn are relavent to the analysis.

Additional comments

This is an interesting assessment, and the use of the haplodiploid genetics of this organism for the work is rather clever. I recommend publication once a more coherent draft is submitted. The assessment needs a thorough review for writing and clarity. The Discussion does a relatively good job of explaining the significance of the paper, whereas the methods and results are unclear especially when addressing the different crosses. The Introduction, although relatively clear, is awkwardly worded at points and should be reviewed thoroughly. The Summary is fine. I cannot necessarily comment on the whole genome amplification method as I have not used it, but from what I can tell it seems like a very interesting method that was used appropriately.

·

Basic reporting

An excellent, unambiguously written article. I have made minor corrections to some of the English early in the article.

I suggest one major, and one minor issue:

1. Major issue: The identity of the origibal culture as T. brassicae needs to be unambiguously established. This does not of course affect the findings of the paper, but it would make it much more useful if the species could be unequivocally named.

This can be done by using the Polaszek et al and the Pino-Perez et al papers cited in the attachment (using ITS 2 sequencing which the Rasplus lab can easily provide). A morphological confirmation using dissected male genitalia would also be useful. I would be happy to provide these details.

2. Minor issue: A recent paper dealing with the use of Trichogramma to control the notorious tomato pest Tuta absoluta (and the banana pest Chrysodeixis chalcites) could be included as a useful reference.

Experimental design

While not an expert on the experimental design area of this paper, I am very familiar with the track records of the 1st and last authors, and am certain of the accuracy and reliability of their protocols. The work is well within the journal's scope.

Validity of the findings

Clearly robust, and most importantly highly innovative findings. The results provide protocols for future work on minute organisms.

Additional comments

Please see general comments above and in the attachment.

---

## Round 0.2 · accepted · Accept

Thanks for your revision of the manuscript and attention to the feedback of the referees. Your revisions provide sufficient detail for the referees and I to follow your experimental design, and I am happy to move it forward into production.

#